# Upcycling of Waste Cherries Produces an Anthocyanin-Rich Powder That Protects Against Amyloid-β Toxicity in *C. elegans*

**DOI:** 10.3390/antiox14080995

**Published:** 2025-08-13

**Authors:** Sarah A. Blackburn, William G. Sullivan, Laura M. Freeman, Kevin Howland, Antonis A. Karamalegos, Michael Dallaway, Mark Philo, Jennifer M. A. Tullet, Marina Ezcurra

**Affiliations:** 1School of Natural Sciences, University of Kent, Canterbury, Kent CT2 7NZ, UK; s.a.blackburn@kent.ac.uk (S.A.B.); wgs3@kent.ac.uk (W.G.S.); lmf42@kent.ac.uk (L.M.F.); k.howland@kent.ac.uk (K.H.); aak31@kent.ac.uk (A.A.K.); 2Rent A Cherry Tree, Cooks Yard Farm, New Road, Northiam TN31 6HS, UK; michael@rentacherrytree.co.uk; 3Science Operations, Quadram Institute Bioscience, Norwich Research Park, Norwich NR4 7UG, UK; mark.philo@quardram.ac.uk

**Keywords:** cherries, anthocyanins, *C. elegans*, bioactives, health, Alzheimer’s disease, upcycling, agricultural waste

## Abstract

Agricultural waste poses significant environmental and economic challenges, with the UK generating 135,000 tonnes annually. Upcycling plant-derived waste offers a sustainable approach to enhancing agricultural productivity while producing innovative, health-promoting foods. Cherries, particularly rich in anthocyanins and quercetin, possess antioxidant and anti-inflammatory properties linked to numerous health benefits. In this study researchers and a small agricultural business in Kent, the UK’s largest cherry-producing region, collaborated to quantify the bioactive compounds in products derived from waste cherries and evaluate their health potential. We find that cherry juice, pulp, and pomace retain high anthocyanin content, particularly Cyanidin-3-O-rutinoside, and contain quercetin. Using *Caenorhabditis elegans* as a model, we demonstrate that cherry pulp supplementation is protective in an Alzheimer’s disease model. Our study highlights the potential to upcycle agricultural waste to produce foods with health benefits while reducing waste.

## 1. Introduction

Waste from agricultural activities leads to reduced productivity and damage to the environment. Globally, over one-third of all food produced, about 3 million tonnes, goes to waste. Fruit and vegetables have the highest wastage rates, often going to waste before even leaving the farm [1]. Plants, including those grown as crops, produce a wider variety of bioactive compounds, many of which have beneficial health effects. However, when plants are harvested for market, the processing often produces waste. This plant-derived waste also remains high in health-promoting bioactive compounds, such as polyphenols [2], meaning that upcycling plant waste holds the potential to drive the production of innovative, inexpensive, sustainable, and healthy foods.

Polyphenols are phytochemicals produced by plants and found in many fruits and vegetables. In plants, polyphenols play crucial roles, acting as antioxidants to quench free radicals, defending against infection by pathogens, and absorbing ultraviolet light to protect tissues from damage [3]. Polyphenols are consumed by humans and animals through the diet, and dietary polyphenols are gaining scientific and consumer attention due to their health benefits. Studies have found that polyphenols have protective effects against cancer, cardiovascular disease, and neurodegenerative diseases [4,5]. Anthocyanins are a subclass of polyphenols acting as pigments that give plants vibrant colours (purple, blue, and red), attracting pollinators and known for their antioxidant, anti-inflammatory, and cardiovascular benefits. Their chemical structure means that they can act as free radical scavengers against harmful oxidants such as reactive oxygen and nitrogen species, quenching reactive radical species by single electron transfer reaction and through hydrogen atom abstraction from phenolic groups [5]. Epidemiological studies have long suggested that diets rich in anthocyanins and other polyphenols slow age-related cognitive decline and reduce the risk of dementia. Thus, consumption of anthocyanin-rich foods could play a role in cognitive health and disease prevention [6,7]. Major dietary sources of anthocyanins include dark-coloured fruits, vegetables, grains, and legumes, e.g., blackcurrants, blackberries, and blueberries (300–500 mg/100 g of fresh product), red cabbage (300–500 mg/100 g), and black rice (200–400 mg/100 g). Cherries also contain relatively high levels of anthocyanins (50–100 mg/100 g) and are widely popular, making them an important source of anthocyanins [5,8,9].

Cherry production is an important horticultural industry globally, with the major producing countries being Turkey, the United States, Chile, and Uzbekistan. In 2022, the estimated total world production was 4.36 million metric tonnes [10]. Cherry production generates a significant amount of agricultural waste, including fruit that is discarded due to it not being marketable. Most markets require fruits that are of uniform size, shape, colour, and ripeness, and fruits that are not within these standards are typically transported to landfills and disposed of. Although there are variations between regions and market standards, it is estimated that approximately 10% of cherries are discarded before reaching consumers due to not meeting quality standards, negatively impacting productivity, carbon emissions, and sustainability [11]. Although cosmetically unattractive, waste cherries are a potential source of natural compounds with beneficial effects on health. Converting cherry waste into food products could lead to increased agricultural sustainability and productivity while at the same time increasing the availability of inexpensive, healthy foods.

Kent, in the southeast of England (Figure 1A), is the largest cherry producer in the UK and generates large amounts of waste cherries. A total of 60 tonnes of Kent cherries that are bruised, soft, or discoloured are transported to landfill every year (Michael Dallaway, personal communication). These cherries have the potential to be developed into new innovative foods with beneficial health and environmental profiles. In this study, researchers in Kent collaborated with a small local horticultural business to analyse anthocyanin content and health properties in products made from cherry waste (Figure 1B,C). Our goal was to demonstrate the value of cherry waste as a means to transform the waste into new foods. Here, we quantify the anthocyanin content of a cherry juice made by pressing waste cherries, as well as the pulp and pomace remaining after juice pressing (Figure 1B,C). We also analyse the content of quercetin, a polyphenol in the flavanol subclass, which, like anthocyanins, is associated with health benefits [12]. We find that all these cherry products are rich sources of several different anthocyanins, including Cyanidin-3-O-rutinoside, and have low, but detectable, levels of quercetin. To examine the importance of these bioactive-containing products to organismal health, we used the model organism *Caenorhabditis elegans*, a valuable tool for studies of health outcomes in vivo (Figure 1C). *C. elegans* offers unique advantages as a whole-organism model for evaluating dietary bioactives, enabling assessment of organism-level responses including immunity, metabolism, and ageing. *C. elegans* is amenable to rapid, high-throughput screening without carrying the ethical or cost limitations associated with vertebrate models, making it ideal for bioactivity screening [13]. We find that supplementing with cherry pulp powder, a product high in anthocyanins, is protective in a *C. elegans* model of Alzheimer’s disease pathology.

Our work shows a high anthocyanin content of bioactives in a specific stream of cherry waste products and demonstrates its potential health benefits. The study provides scientific evidence for innovative approaches that can be adopted at scale in Kent and other agricultural regions. and generates information relevant to consumers, markets, and growers.

## 2. Materials and Methods

### 2.1. Ultra-Performance Liquid Chromatography (UPLC-MS) Coupled with Quadrupole Time-of-Flight Mass Spectrometry (UPLC-QToF)

Samples were analysed by UPLC-MS to identify target anthocyanins, then quantified using triple quadrupole. Liquid samples were analysed directly, and solid samples were extracted with solvent prior to analysis. Liquid samples were additionally diluted with extraction solvent 10-fold to account for high-concentration compounds.

#### 2.1.1. Sample Preparation

For comparison of cherry juice batches, cherry juice was centrifuged at 13,200 RPM (14,220*× g*) for 10 min. Supernatants were then filtered (0.2 µm pore size filter) and stored at −80 °C.

#### 2.1.2. Ethanol Extraction

Whole cherry, cherry pulp, and cherry pomace samples were extracted using acidified aqueous ethanol (50% ethanol with 0.5% HCl, *v*/*v*) at a solvent-to-sample ratio of 5:1 (mL/g). The mixtures were incubated for 24 h at 4 °C on a roller mixer and then centrifuged at 13,200 RPM (14,220× *g*) for 10 min. Supernatants were then filtered (0.2 µm pore size filter) and stored at −80 °C. Cherry juice was freeze-dried for 24 h to produce a solid that could be used for direct comparison by ethanol extraction.

#### 2.1.3. Standards

Individual standards were diluted with formic acid/methanol/water (1/14/35 *v*/*v*/*v*) such that the calibration range of the anthocyanin was at 50, 10, 5, 2, 1, 0.5, and 0.2 µg/mL in solution. Quercetin was prepared at 10–0.2 µg/mL. Samples were stored at −20 °C until analysis. Cyanidin 3-o-glucoside standard was from Extrasynthase, Genay, France (code 0915 S). Quercetin (95%) was from Merck, Darmstadt, Germany (code Q4951). Anthocyanins were measured against a cyanidin-3-O-glucoside standard and reported as equivalent concentrations

#### 2.1.4. UPLC-QToF

Instrumental analysis was made using a Waters Acquity UPLC coupled to a Select Series Cyclic QToF mass spectrometer (Wilmslow, UK). The analytical column was a Waters HSS T3 C18 100 × 2.1 mm × 1.7 µm at 35 °C. Mobile phases were solvent A: water with 1% formic acid and solvent B: acetonitrile with 1% formic acid with a constant flow of 0.4 mL/min. A gradient was applied as 5% B at injection for 1 min, 10% B at 5 min, 16% B at 15 min, 95% B at 40 min, and then back to initial conditions for 3 min. Sample injections were 2 µL. To identify each anthocyanin, the instrument was operated in positive electrospray ionisation mode in full scan to determine the molecular ion (low fragmentation energy) and fragments (high fragmentation energy), particularly the associated anthocyanin aglycone fragment. The identified anthocyanins were then analysed by a triple quadrupole instrument, looking at the molecular ion and fragment ion only for selectivity and sensitivity.

#### 2.1.5. UPLC-MS/MS

Instrumental analysis was made using a Waters Acquity UPLC coupled to a TQ-Absolute triple quadrupole mass spectrometer. The analytical column was a Waters HSS T3 C18 100 × 2.1 mm × 1.7 µm at 35 °C. Mobile phases were solvent A: water with 1% formic acid and solvent B: acetonitrile with 1% formic acid with a constant flow of 0.4 mL/min. A gradient was applied as 5% B at injection for 1 min, 10% B at 5 min, 16% B at 15 min, 95% B at 16, and then held for 2 min, then back to initial conditions for 3 min. Sample injections were 1 µL.

### 2.2. Preparation of Cherry Product Stock Solutions for Bioactivity Assays

#### 2.2.1. Cherry Juice

Dallaways’ Kentish Cherry Juice was freeze-dried for 48 h to remove all moisture. The residue was resuspended in 100% DMSO at a 1 g:2 mL ratio, vortexed, and then incubated with rocking at room temperature for 1 h to resuspend. The sample was centrifuged at 14,220*× g* for 10 min. Supernatant was taken and filtered (0.2 µm pore), split into aliquots, and stored at −20 °C.

#### 2.2.2. Cherry Pulp Powder

Cherry pulp powder was produced using cherry waste from the Dallaways juice pressing process. Waste cherry pulp was filtered using a pulping machine, then freeze-dried and milled into a fine powder. The powder was resuspended in 100% DMSO at a 1 g:4 mL ratio, vortexed, and then incubated with rocking at room temperature for 1 h to resuspend. The sample was centrifuged at 14,220*× g* for 10 min. Supernatant was taken and filtered (0.2 µm pore), split into aliquots, and then stored at −20 °C.

#### 2.2.3. Blackberry Powder

Blackberry extract powder was sourced from Natural Products of Worth (NPOW). The powder was resuspended in 100% DMSO at a 1 g:20 mL ratio, vortexed, and incubated with rocking at room temperature for 1 h to resuspend. The sample was centrifuged at 14,220*× g* for 10 min. The supernatant was taken and filtered (0.2 µm pore size), split into aliquots, and then stored at −20 °C.

#### 2.2.4. Kirby–Bauer Analysis

Bacterial cultures of *E. coli* OP50, *E. coli* K-12, *Staphylococcus albus*, *Pseudomonas aeruginosa* PA14, and *Bacillus megaterium* were inoculated into LB and incubated overnight at 37 °C (30 °C for *B. megaterium*) with shaking at 180 RPM. The optical density at 600 nm (OD_600_) of each culture was measured and diluted to 0.1 using LB. A total of 100 µL was evenly spread onto LB agar plates, which were left to dry at room temperature for 30 min. Sterile filter paper discs were treated with 15 µL of either anthocyanin solution (in DMSO), a negative control (sterile Milli-Q^®^ water or DMSO), or a positive control (50 µg/mL kanamycin or 100 µg/mL ampicillin) and then left to dry for a minimum of 2 h. Discs were placed onto the relevant bacterial lawns and incubated at 37 °C (30 °C for *B. megaterium*) for 24 h. The diameter of the zones of inhibition was measured in mm.

#### 2.2.5. Growth Measurements of *E. coli* OP50

*E. coli* OP50 was inoculated into LB and incubated overnight at 37 °C with shaking at 180 RPM. The overnight culture was diluted 1000× in LB. A sterile 96-well plate was set up to include 100 µL per well of the following negative controls diluted in LB: (1) 1% DMSO, (2) anthocyanins only, (3) OP50 bacteria only, and (4) OP50 with 1% DMSO. OP50 bacteria were grown in the presence of anthocyanins of varying concentrations. The relevant values for the ‘Anthocyanin only’ control were subtracted from each OP50 growth curve to account for the strong pink/red colour of anthocyanins. A total of 100 µL total volume was added to each well of a sterile 96-well plate, and each condition was performed in triplicate to give 3 technical repeats per plate. The OD600 was measured using a BMG Labtech SPECTROstar Nano plate reader, where the plate was incubated at 37 °C for a minimum of 18 h with shaking at 400 RPM. Each experiment was also repeated to give 3 biological repeats.

#### 2.2.6. Growth Measurement of *Lactobacillus* Strains

The following *Lactobacillus* strains were sourced from the DSMZ (German Collection of Microorganisms and Cell Cultures): *L. acidophilus* (#20079), *L. brevis* (#20054), *L. helveticus* (#20075), *L. paracasei* (#28675), *L. plantarum* (#20174), *L. rhamnosus* (#111733), and *L. salivarius* (#20555). All strains were streaked onto MRS (Fisher Scientific #11713553) agar plates and incubated at 37 °C for 48 h with 5% CO_2_. Colonies were inoculated into MRS liquid media and incubated at 37 °C with 5% CO_2_ for 48 h. For growth of *L. acidophilus* and *L. helveticus*, the MRS media was supplemented with 0.05% cysteine hydrochloride. The overnight cultures were diluted 1000× in MRS (+/− 0.05% cysteine hydrochloride). Sterile 96-well plates were set up to include 100 µL per well of the following negative controls diluted in MRS: (1) 1% DMSO and (2) anthocyanins only. Bacteria were grown in the presence of either 1% DMSO or 100 µg/mL of the relevant anthocyanin. The values for the ‘Anthocyanin only’ controls were subtracted from each bacterial growth curve to account for the strong pink/red colour of anthocyanins. A total of 100 µL total volume was added to each well of a sterile 96-well plate, and each condition was performed in triplicate to give 3 technical repeats per plate. The OD600 was measured using a BMG Labtech SPECTROstar Nano plate reader, where the plate was incubated at 37 °C for a minimum of 48 h with shaking at 400 RPM. Each experiment was also repeated to give 3 biological repeats.

The maximum rate of growth was calculated by subtracting the average values of the sample media controls from each technical replicate of the growth data, then plotting the graph of each subtracted technical replicate density against time. The exponential phase of the growth curve was used to calculate the maximum rate of growth of the given bacteria, OD600/hour. The maximum growth rate was then standardised to the respective solvent control.

#### 2.2.7. *C. elegans* Culture Methods and Strains

*C. elegans* were maintained at 20 °C on Nematode Growth Medium (NGM) plates seeded with *Escherichia coli* OP50. *C. elegans* strains used in this study were N2 (wild-type), GMC101 *dvIs100 [unc-54p::A-beta-1-42::unc-54 3′-UTR + mtl-2p::GFP]* and CL2166 *dvIs[(pAF15)gst-4p::GFP::NLS]*.

#### 2.2.8. Developmental Assays

Fifteen adult *C. elegans* were placed on experimental plates and allowed to lay eggs for 2 h. After 68 h, the number of animals at L4 and adult stages was counted to evaluate effects on developmental rate. The experiment was repeated three times.

#### 2.2.9. Proteotoxicity Assay

GMC101 animals were allowed to lay eggs for 4 h on supplemented plates to synchronise the population. ~56 h after the egg lay, L4 stage animals were transferred to new plates and shifted to 25 °C to induce expression of amyloid-β_42_. Animals were scored every day for four days, and the numbers of paralysed and non-paralysed animals were counted. *C. elegans* were counted as paralysed if they were alive but failed to move forwards or backwards after stimulation with a platinum wire. Three biological replicates were performed with three plates containing 30–40 animals each per condition.

#### 2.2.10. Imaging of GST-4::GFP

Day 1 adults were mounted on 2.5% agarose pads and anaesthetised with 25 mM tetramisole. Fluorescence microscopy was performed using a Leica DMR compound epifluorescence microscope (filter cubes: GFP, λ*ex* 450/90 nm, λ*em* 500/50 nm) fitted with a Leica DFC camera. All images were captured using the software Micromanager 2.0. Imaging was performed with a 10× objective and an exposure time of 10 ms for brightfield and 300 ms for fluorescence images. Image analysis was performed using Fiji ImageJ v. 2.15.1 by measuring the mean grey area within the entire body of each worm. Three biological replicates were performed with a minimum of 22 worms per condition and replicate.

#### 2.2.11. Statistical Analysis

Data processing and statistical evaluation were performed with GraphPad Prism, version 10.1.1 (270) (San Diego, CA, USA).

## 3. Results

### 3.1. Quantification and Identification of Anthocyanins in Kent Cherry Products

Juicing cherries is an economically viable way of using up fruit that is not sold directly to consumers. As it is reported that cherries have high levels of anthocyanins [5,8], we determined the levels of anthocyanins in cherry juice made from waste cherries. We first quantified total anthocyanin content in juice made from three independent harvests (2021, 2022, and 2023). For all batches, the juice was bottled straight after pressing and stored identically until analysis. Juice samples were compared to a known panel of standards using Liquid Chromatography and Mass Spectrometry (LC-MS). This identified high levels of total anthocyanins in all samples, ranging from 42 to 110 μg/mL (Figure 2A), with the 2023 batch having the highest anthocyanin content. Anthocyanin levels in plants are increased by high temperatures and exposure to sunlight and degrade at room temperature [14]; thus, the higher anthocyanin content in the 2023 batch is likely either a result of a shorter storage time or the higher than average temperatures in 2023 (e.g., record-breaking temperatures in June 2023) [15].

Waste from the production of cherry juice was taken and processed through a pulping machine to give cherry pulp and pomace samples (Figure 1B). To determine whether these by-products contained valuable bioactives, we examined total anthocyanin levels in pulp and pomace produced from juice pressing of the 2022 and 2023 cherry harvests. We also generated two new products, powders made from cherry pulp from the 2022 harvest. The powders were made by either freeze-drying or air-drying pulp and then milling it into a fine powder. Anthocyanin levels in these products were compared to whole fresh cherries and the cherry juice. In addition, we included a commercially available blackberry extract powder as a comparison. The samples were prepared using ethanol-based extraction with acidified conditions [16] and analysed by LCMS. We found that while whole cherries contained a total of 431 μg/g anthocyanins, levels in cherry juice were higher at 597 μg/g (Figure 2B). Cherry pulp contained 5 μg/g total anthocyanin, and pomace contained 21 μg/g, likely as a result of pomace containing the skin, which is where most of the anthocyanins reside. The air-dried pulp powder had a lower anthocyanin content of 7.5 μg/g, while the freeze-dried pulp powder had a relatively high content of 462 μg/g (Figure 2B). This is consistent with other studies showing improved retention of anthocyanins in freeze-dried fruits compared to hot-air drying due to instability at higher temperatures [17]. The levels of anthocyanins in our samples were comparable to a commercially available blackberry powder enriched with anthocyanin extracts, which contained 1864 μg/g anthocyanins (Figure 2C) [10,18].

Next, we identified and quantified the anthocyanin species present in the samples. We detected a mixture of eight different anthocyanins, with Cyanidin-3-O-rutinoside being the predominant species across all cherry harvests and sample types, constituting 69–87% of total anthocyanins (Table 1 and Table 2). In contrast, in the blackberry powder, Cyanidin-3-glucoside was the main anthocyanin component (Table 3). Taken together, these data show that cherry juice made from waste fruit is rich in anthocyanins and that high levels of anthocyanins remain in the leftover pomace and pulp and in powder made of freeze-dried pulp. Due to the negligible amounts of anthocyanins in air-dried pulp powder, we only proceeded with the freeze-dried pulp powder (Figure 2C).

In addition to anthocyanins, cherries also contain quercetin, a flavonoid with antioxidant and antibacterial functions. To determine the presence and concentration of quercetin in Kent cherries, we analysed our samples using LC-MS against quercetin standards. As for the anthocyanins, we detected a higher quercetin content in the juice from the 2023 batch compared to the 2021 and 2022 batches (8-, 3-, and 27-fold higher levels, respectively; Figure 2D). In comparison to anthocyanins, overall quercetin content was low, ranging from 0.04 to 1.18 mg/L. Our analyses showed a high degree of variation in quercetin across cherry juice batches. This likely reflects year-to-year differences in growing conditions such as sunlight, temperature, and water availability, all of which are known to influence flavonoid biosynthesis [14,15].

### 3.2. Anti-Bacterial and Prebiotic Properties of Cherry Juice and Cherry Pulp Powder

Anthocyanins have been shown to alter bacterial growth [17]. Biologically, this would be beneficial to health if they inhibit the growth of pathogenic bacteria or promote the growth of probiotic bacteria. Given the levels of anthocyanins and quercetin identified in the cherry products (Figure 2A–D), we assessed their impact on bacterial growth. First we tested their antimicrobial properties using the Kirby–Bauer test, a microbiological assay used to determine how sensitive bacteria are to antimicrobials [18]. We first determined how 100 μg/mL of individual anthocyanins (Cyanidin-3-O-rutinoside, Cyanidin-3-glucoside, Pelargonidin-3-O-rutinoside, Peonidin-3-O-rutinoside) impacted the growth of the Gram-negative bacteria *E. coli OP50* and *K-12* and the Gram-positive bacteria *S. albus* and *B. megaterium* on agar. While kanamycin control (50 μg/mL) resulted in inhibition zones ranging between 12.5 and 19.0 mm for the different bacteria, none of the anthocyanins impacted bacterial growth. In contrast, 100 μg/mL quercetin generated inhibition zones of 9.0 mm for both *S. albus* and *B. megaterium*, confirming the antimicrobial properties of this compound. (Figure 3A). We also quantified the ability of the cherry juice and cherry pulp powder to inhibit the growth of bacteria. The DMSO stocks of cherry juice, pulp powder, and blackberry extracts were also added to filter discs but it was found that neither of the cherry product samples nor the blackberry extract inhibited bacterial growth (Figure 3B).

Next, we tested whether the same samples have prebiotic properties by conducting bacterial growth curve assays. Seven probiotic strains from the *Lactobacillus* genus (*L. brevis*, *L. paracasei*, *L. plantarum*, *L. helveticus*, *L. salivarus*, *L. acidophilus*, and *L. rhamnosus*) were grown for 48 hrs with and without the cherry products, individual anthocyanins, or controls, and their growth was quantified using OD_600_ measurements. For *L. salivarus*, we found that treatment with the pulp powder resulted in an 8.6% increase in maximum growth rate (Figure 3C). In contrast, the pulp powder, as well as the blackberry powder, had a strong inhibitory effect on *L. acidophilus*, reducing maximum growth rate by 48% and 52%, respectively (Figure 3D). Growth of *L. rhamnosus* was mildly inhibited by Cyanidin-3-O-rutinoside, Cyanidin-3-glucoside, and cherry juice (7–16%, Figure 3D), and by 22% by the blackberry powder. Treatment with cherry products or anthocyanins did not affect growth of *L. brevis*, *L. paracasei*, *L. plantarum*, or *L. helveticus* (Appendix A). Quercetin at 100 μg/mL inhibited *L. rhamnosus* (Figure 3C) and *L. paracasei* (Appendix A). Except for the slight beneficial effect of the pulp powder on the growth of *L. salivarus*, our findings are not consistent with cherry products or individual anthocyanins having probiotic properties.

### 3.3. Cherry Products Do Not Have a Negative Impact on C. elegans Growth or Development

We next asked whether the cherry products impact organismal health in the whole-organism model *C. elegans*. The ease of cultivation, rapid generation time, genetic tractability, and extensive toolbox of *C. elegans* enable high-throughput testing to determine bioactivity across a wide range of health outcomes [13]. *C. elegans* is a bacterivore and, under laboratory conditions, feeds on *E. coli* OP50. To rule out the possibility that cherry products inhibit the growth of OP50 and thereby impact food availability or quality, we quantified OP50 growth in the absence and presence of individual anthocyanins, cherry juice, or pulp powder. We found that both the cherry juice and cherry pulp powder increased bacterial growth, increasing the maximum growth rate and the final culture density. In contrast, blackberry powder reduced the growth rate of OP50 (Figure 4A and Appendix A). Treatment with individual anthocyanins at 100 µg/mL did not alter maximum growth rate, while quercetin resulted in a small but significant decrease in maximum growth rate (Figure 4B,C and Appendix A).

This demonstrates that none of the samples inhibit growth of OP50, and supplementing the *C. elegans* OP50 growth plates with them will not negatively impact their nutrient source. Some plant compounds can be toxic, so we next asked if the cherry samples have detrimental effects on *C. elegans* growth and development, both of which are good readouts of overall health. After hatching, *C. elegans* pass through four larval stages before becoming reproductively capable adults (Figure 4D). We measured the time it took for *C. elegans* treated with cherry products, blackberry powder, individual anthocyanins (100 µg/mL) or quercetin (10 µg/mL and 100 µg mg/mL) to pass through larval development and reach adulthood compared to controls. Consistent with other studies [19], the solvent control delayed development. We found that after 68 h, ~60% of *C. elegans* treated with the solvent control reached adulthood compared to ~80% of animals on OP50 only (Figure 4E,F). However, the addition of juice, pulp powder, blackberry extract, or individual compounds did not impact *C. elegans* development (Figure 4E,F). Together our data shows that the cherry products do not alter *C. elegans* growth, suggesting that they are non-toxic and making this model a viable option for conducting bioactivity assays.

### 3.4. Cherry Pulp Powder Protects Against Amyloid-β Proteotoxicity

Anthocyanins have been reported to protect against neurodegenerative diseases characterised by the progressive loss of neurons and the accumulation of misfolded proteins in the brain, improving cognition and neuroprotection [5,20]. Given their chemical composition, we reasoned that the cherry products tested here may also have similar effects. To test this, we utilised a humanised *C. elegans* model of proteotoxicity [21]. In this model, human amyloid-β_42_ is expressed in the body-wall muscle, leading to aggregation of amyloid-β_42_ in muscle, ultimately resulting in muscle dysfunction and paralysis of the animal and enabling quantification of proteotoxicity by inspection using a dissection microscope (Figure 4G). We evaluated proteotoxicity in this model in normal conditions and in the presence of cherry products and blackberry extract by measuring paralysis on day 2 of adulthood in amyloid-β_42_-expressing animals. As previously shown, under normal conditions 95% of *C. elegans* expressing the amyloid are paralysed within 2 days (Figure 4H). Supplementation with cherry pulp powder had a protective effect, reducing paralysis by 28% compared to controls, while blackberry extract was not protective (Figure 4H). Purified anthocyanin supplements are popular health foods. However, treating *C. elegans* with individual anthocyanins had no impact on amyloid-induced pathology (Figure 4I). We also tested supplementation with quercetin at 10 µg/mL and 100 µg/mL but did not observe any effects (Figure 4J). Our data showing that the cherry pulp powder, but not the cherry juice, blackberry powder, individual anthocyanins, nor quercetin, exhibited protective effects were unexpected since all products contain high levels of anthocyanins. These findings raise the possibility that the observed protection against proteotoxicity may not be mediated via an anthocyanin-induced oxidative stress response. To examine the effect of the products on oxidative stress pathway activation, we utilised a well-established Nrf-2/SKN-1-responsive reporter strain, *gst-4p::GFP*, which enables in vivo monitoring of Nrf-2/SKN-1 activation in response to oxidative stress and dietary interventions using fluorescence microscopy [22]. We found that neither cherry juice, blackberry powder, nor cherry pulp powder alters *gst-4::GFP* levels relative to vehicle control. In contrast, 1 mM paraquat, a known pro-oxidant, robustly increased *gst-4*::GFP levels, confirming the assay’s responsiveness (Appendix A). These results suggest that the protective effects of cherry pulp powder in the Aβ model are unlikely to be mediated through Nrf2/SKN-1-driven oxidative stress response pathways. A possibility is that other constituents in the pulp powder, or interactions that enhance the bioactivity of anthocyanins and quercetin, may contribute to the observed protective effects, and that more complex interactions between bioactive compounds in a food matrix are required to improve health.

## 4. Discussion

### 4.1. Insights from Analysing Waste Cherry Products

Here, we describe the anthocyanin content and bioactive properties of products made from cherry waste. Our findings demonstrate high concentrations of anthocyanins in juice, pulp, and pulp powder made from cherries that would otherwise be destined for landfill. As anthocyanins have established health-promoting properties, these findings strongly suggest that food products made from horticultural waste can be developed into inexpensive, health-promoting foods. Our study also demonstrates the beneficial bioactivity of products made from waste and the benefits of working locally to improve grower-researcher collaborations.

Using a *C. elegans* model, we showed that our upcycled cherry pulp powder protects against Alzheimer’s-related Aβ toxicity in a *C. elegans* model, implying neuroprotective effects. It is important to note that these results should be interpreted in light of certain limitations. Although *C. elegans* models conserved molecular pathways, the findings might not directly translate to human biology. Additionally, we did not assess anthocyanin bioavailability, an important factor for determining efficacy. However, clinical trials have shown that anthocyanin-rich extracts have neuroprotective effects across age groups. Dietary supplementation with extracts from blueberries improved cognitive performance in older adults [23], in elderly adults with cognitive impairment [24], in healthy adults [25,26,27], and in children [28]. Less is known about using cherry-based supplements, but consumption of anthocyanin-rich cherry juice for 12 weeks has been shown to improve memory and cognition in older adults with mild-to-moderate dementia [29]. Future research will determine whether cherry products, like blueberries, also have broad effects on cognitive function and have the potential to prevent or delay AD and dementia.

Several studies have shown that anthocyanins have prebiotic properties, enhancing/promoting growth of gut bacteria, and bactericidal properties, inhibiting the growth of pathogens. In our study we did not find any positive nor negative effects on growth of *Lactobacilli* probiotic strains. We also did not observe antibiotic effects when conducting tests on pathogenic bacteria. Differences between our findings and other studies could be due to the use of different experimental approaches, e.g., in vitro or in vivo assays, oxygen concentrations, and experimental conditions, e.g., solid media or liquid media, as well as the bacterial strains that were tested.

### 4.2. Mechanisms by Which Cherry Powder May Protect Against Aβ Toxicity

Our finding that cherry powder protects against Aβ-associated toxicity in a *C. elegans* Aβ model is consistent with work in other experimental models. Several studies in mouse models show that anthocyanin supplementation has significant potential to prevent AD [20]. For example, anthocyanin-enriched bilberry and blackcurrant extracts reduce Aβ production, Aβ deposition, Aβ-associated memory loss, and neurodegeneration in transgenic mice expressing human Aβ [30,31,32]. Bilberry-derived anthocyanin mixtures inhibit formation of Aβ peptide fibrils in cell-free in vitro experiments and reduce Aβ toxicity in neuronal cells [32]. The anthocyanin Cyanidin-3-O-glucoside, found at high levels in blackberries and blueberries, can inhibit aggregation of Aβ into oligomers and Aβ neurotoxicity in human neuronal SH-SY5Y cells [33]. These positive effects may be due to the antioxidant or anti-inflammatory activity of anthocyanins [5,8,20], but also via anthocyanins directly interfering with Aβ peptide oligomerisation and preventing the formation of toxic fibrils [32].

In our study, we found beneficial effects from supplementation with cherry pulp powder, but not from cherry juice, which has a similar anthocyanin profile as the powder. The blackberry extract, which has a high anthocyanin content but a different anthocyanin profile, did not show bioactivity either, nor did supplementation with individual anthocyanins. The bioactive properties of the cherry powder could be a result of the precise anthocyanin composition of the product, differences in other components, or the food matrix of the powder. The food matrix refers to the intricate interactions between nutrients, fibre, bioactive compounds, and other components that play a crucial role in shaping how a food is digested and metabolised and its beneficial effects upon consumption [34]. Studies investigating the bioavailability of anthocyanins in a food matrix compared to anthocyanin-rich extracts suggest that food matrices protect anthocyanins during digestion, leading to increased stability and absorption of anthocyanins [35]. Anthocyanins are chemically unstable under neutral or alkaline conditions and are prone to degradation during gastrointestinal transit. The presence of other components in the cherry pulp matrix, such as fibre pectin, and proteins, may act as physical or chemical stabilisers, protecting anthocyanins from early breakdown and enhancing their persistence through the digestive tract. This could help explain the greater biological activity observed with the pulp powder compared to anthocyanin standards or juice alone. Thus, consuming anthocyanin-rich cherry powder might impart health benefits that are not observed in extracts or supplements.

### 4.3. An Opportunity to Use Waste to Produce Inexpensive Healthy Foods

With the global population increasing and demographics rapidly shifting towards ageing societies with increasing numbers of people suffering from multiple age-related chronic diseases, there is a growing demand for the production of foods that meet our growing health needs while being sustainable [36]. Moreover, consumer focus is moving away from foods that simply supply energy and basic nutrition to those that also contain bioactive compounds with health benefits, e.g., disease protection. In some markets, there is also a consumer desire to recycle, reuse, and buy sustainable products. Our study demonstrates that agricultural waste streams can be sources for healthy food products, such as pulp powders. Such powders offer other advantages in addition to health benefits; they are rich in flavour, have long shelf lives, are easy to transport and store, and can be incorporated into a variety of foods. Our data also show that cherry powder can retain high levels of anthocyanins, in particular Cyanidin-3-O-rutinoside. Cyanidin-3-O-rutinoside is concentrated in the skin and outer mesocarp, tissues that remain in all fractions during processing [37], and is glycosylated, improving its ability to withstand processing steps like juicing or freeze-drying [38]. These findings support the potential of cherry waste-derived powders as a viable approach for retaining stable, bioactive compounds and their value in developing functional food ingredients. As agricultural waste is a major contributor to reduced productivity and negative environmental impact, upcycling plant waste in this way has significant potential to yield a supply of inexpensive, sustainable, and healthy foods, increase productivity, and build a circular bioeconomy.

## Figures and Tables

**Figure 1 antioxidants-14-00995-f001:**
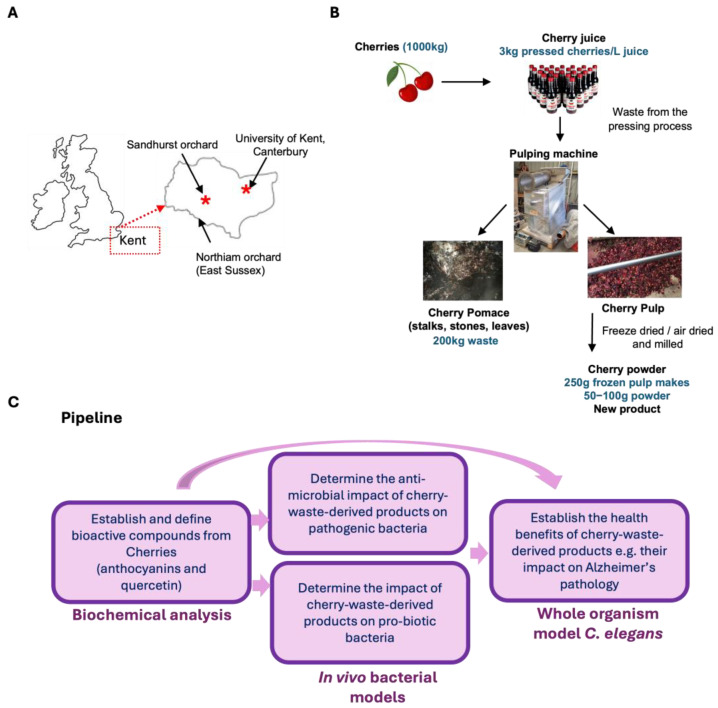
Pipeline for analysing products made from cherry waste. (**A**) Location of cherry orchards and the university campus in Kent, the United Kingdom. (**B**) Fresh (waste) cherries, otherwise destined for landfill, are pressed into juice using a pulping machine. Pulping also produces pomace and pulp, the latter of which is dried and milled into cherry powder. (**C**) Experimental pipeline to establish bioactive constituents and health benefits of cherry products.

**Figure 2 antioxidants-14-00995-f002:**
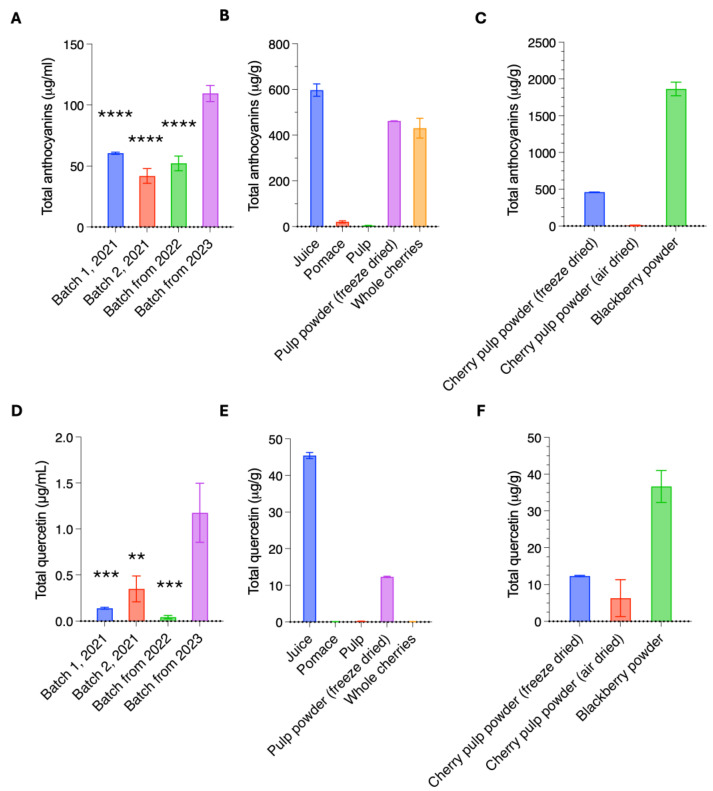
Cherry products contain high levels of bioactive compounds. Total levels of anthocyanins in different batches of cherry juice (**A**) or products (**B**,**C**). Total levels of quercetin in different batches of cherry juice (**D**) or products (**E**,**F**). One representative of three technical replicates is shown for each sample. Error bars represent ± SEM. Statistical analysis was performed using one-way ANOVA, followed by Tukey’s post hoc test for multiple comparisons, comparing with Batch from 2023. ** *p* < 0.01, *** *p* < 0.001, **** *p* < 0.0001.

**Figure 3 antioxidants-14-00995-f003:**
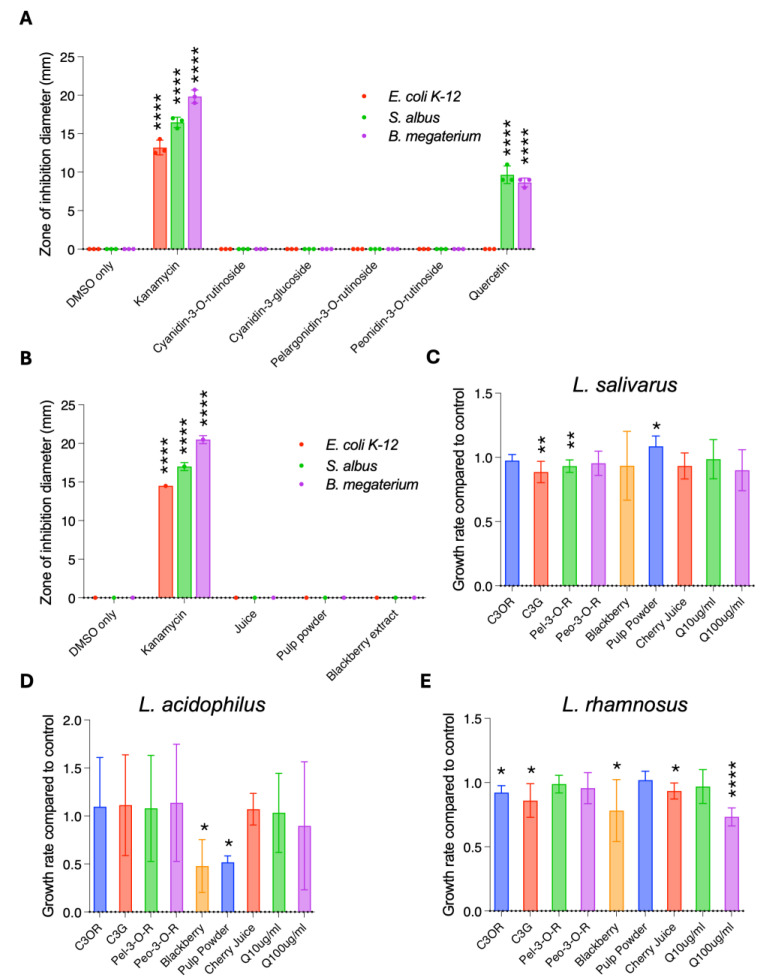
Cherry anthocyanins and products do not impact the growth of common pathogens and probiotics. (**A**,**B**) Kirby–Bauer disc diffusion susceptibility test on different bacterial strains, including common pathogens. Growth of bacteria is inhibited by quercetin but not by anthocyanins (100 µg/mL each) (**A**) or by cherry products (**B**). Statistical analysis by multiple unpaired t-tests, in comparison to DMSO control. (**C**–**E**) Maximum growth rates of probiotic bacteria. Individual anthocyanins at 50 µg/mL or cherry products themselves have minor or no effects on growth rates of *L. rhamnosus* or *L. salivarus* (**C**,**D**). The effect of anthocyanins, cherry products, and quercetin on the growth rate of *L. rhamnosus, L. salivarious*, and *L. acidophilus* is reduced by blackberry powder and cherry pulp powder. (**E**). Statistical analysis by Student’s t-test, two-tailed, homoscedastic. Error bars represent ± SEM. (**A**,**B**) One representative of three technical replicates is shown. (**C**,**D**) combined results from 5–10 biological replicates shown. * *p* < 0.05, ** *p* < 0.01, **** *p* < 0.0001.

**Figure 4 antioxidants-14-00995-f004:**
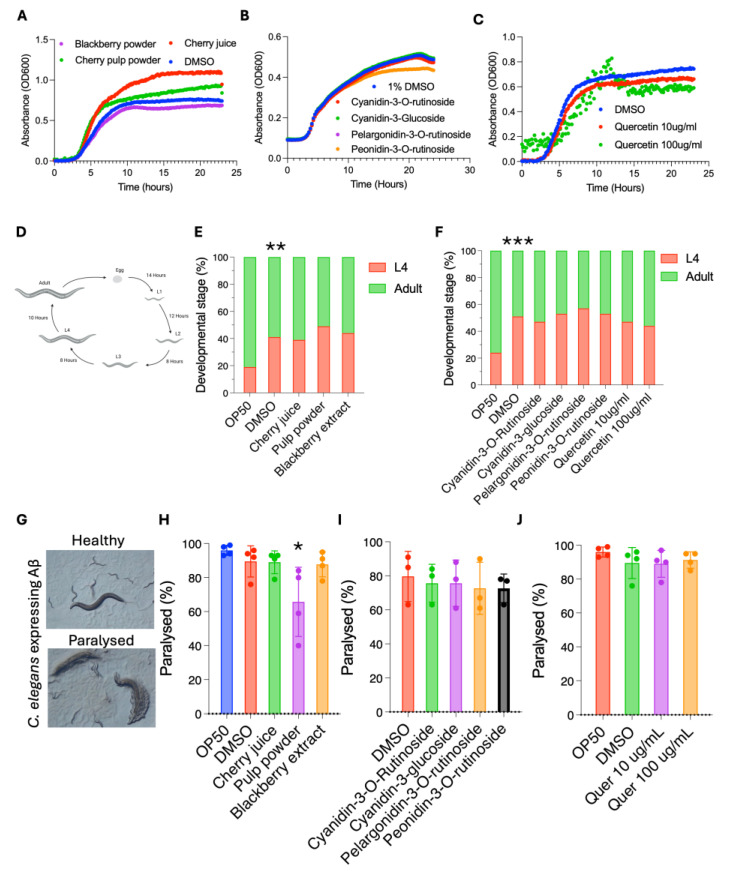
Cherry pulp powder improves health in a *C. elegans* proteotoxicity model of Alzheimer’s disease. (**A**–**C**) Growth of the *C. elegans* laboratory food source *E. coli* OP50. OP50 growth is not altered by cherry products (**A**), individual anthocyanins (**B**), or quercetin (**C**). (**D**) Schematic of *C. elegans* development at 20 °C made in Biorender. (**A**–**C**) n = 9 for each sample. (**E**,**F**) *C. elegans* developmental rate. Development is not impacted by cherry products (**E**), individual anthocyanins, or quercetin (**F**) compared to vehicle controls. The solvent DMSO negatively affects development compared to non-DMSO controls. Statistical analysis by Fisher’s exact test. (**G**) *C. elegans* model Alzheimer’s disease pathology. Human Aβ peptide is expressed in the *C. elegans* body wall muscle, leading to paralysis. (**H**–**J**) Scoring of paralysis in animals supplemented with cherry products, individual anthocyanins, or quercetin. Cherry pulp powder reduces paralysis (**H**). Paralysis is not affected by other products, by anthocyanins (**I**), or quercetin (**J**). (**H**–**J**) Statistical analysis by two-way ANOVA. Combined data from four (**H**,**J**) or three (**I**) biological replicates, each containing 100–120 animals. * *p* < 0.05, **, *p* < 0.01; ***, *p* < 0.001.

**Table 1 antioxidants-14-00995-t001:** Concentration of anthocyanins (mg/L) in different batches of cherry juice.

Sample Name	Biological Replicate	Cyanidin-3-glucoside	Cyanidin-3-O-rutinoside	Pelargonidin-3-glucoside	Pelargonidin-3-O-rutinoside	Delphinidinin-3-glucoside	Delphinidinin-3-rutinoside	Petunidin-3-glucoside	Petunidin-3-rutinoside	Peonidin-3-rutinoside	Peonidin-3-glucoside	Total Anthocyanins
Dallaways cherry juice—Batch 1, 2021	1	5.9	46.6	0.1	0.9	0.0	0.1	0.0	0.0	5.6	0.0	59.2
	2	5.9	46.7	0.1	1.0	0.0	0.1	0.0	0.0	5.9	0.0	59.7
	3	6.2	48.8	0.1	1.0	0.0	0.1	0.0	0.0	6.0	0.0	62.2
Dallaways cherry juice—Batch 2, 2021	1	3.2	44.1	0.1	1.0	0.0	0.1	0.0	0.0	5.4	0.0	53.9
	2	1.9	29.7	0.0	0.7	0.0	0.0	0.0	0.0	3.5	0.0	35.8
	3	1.9	30.0	0.0	0.7	0.0	0.0	0.0	0.0	3.5	0.0	36.1
Dallaways cherry juice—Batch from 2022	1	3.3	50.9	0.1	1.1	0.0	0.1	0.0	0.0	3.7	0.0	59.2
	2	2.0	35.2	0.0	0.7	0.0	0.0	0.0	0.0	2.4	0.0	40.3
	3	3.3	49.1	0.1	1.1	0.0	0.1	0.0	0.0	3.5	0.0	57.2
Dallaways cherry juice—Batch from 2023	1	10.7	71.8	0.2	1.8	0.0	0.1	0.0	0.0	12.3	0.0	96.9
	2	16.2	82.8	0.3	2.5	0.0	0.1	0.0	0.0	17.0	0.0	118.9
	3	15.1	79.5	0.2	2.3	0.0	0.1	0.0	0.0	15.6	0.0	112.8

**Table 2 antioxidants-14-00995-t002:** Concentration of anthocyanins (μg/g) in different cherry sample types.

Sample Name	Biological Replicate Number	Cyanidin-3-glucoside	Cyanidin-3-O-rutinoside	Pelargonidin-3-glucoside	Pelargonidin-3-O-rutinoside	Delphinidinin-3-glucoside	Delphinidinin-3-rutinoside	Petunidin-3-glucoside	Petunidin-3-rutinoside	Peonidin-3-rutinoside	Peonidin-3-glucoside	Total Anthocyanins
Juice	1	91.5	429.0	1.5	14.5	0.0	0.0	0.0	0.0	99.5	0.0	636.0
	2	64.5	392.5	1.0	11.0	0.0	0.0	0.0	0.0	75.5	0.0	544.5
	3	86.0	420.5	1.5	13.0	0.0	0.0	0.0	0.0	90.0	0.0	611.0
Pomace	1	2.0	20.0	0.0	1.0	0.0	0.0	0.0	0.0	3.0	0.0	26.0
	2	2.0	19.5	0.0	1.0	0.0	0.0	0.0	0.0	3.0	0.0	25.5
	3	1.0	8.5	0.0	0.5	0.0	0.0	0.0	0.0	1.5	0.0	11.5
Pulp	1	0.5	3.5	0.0	0.0	0.0	0.0	0.0	0.0	0.5	0.0	4.5
	2	0.5	4.0	0.0	0.0	0.0	0.0	0.0	0.0	0.5	0.0	5.0
	3	0.5	5.0	0.0	0.0	0.0	0.0	0.0	0.0	0.5	0.0	6.0
Pulp powder(freeze dried)	1	28.0	399.5	0.5	7.0	0.0	0.0	0.0	0.0	25.5	0.0	460.5
	2	28.0	401.5	0.5	7.0	0.0	0.0	0.0	0.0	26.0	0.0	463.0
	3	28.5	401.0	0.5	7.0	0.0	0.0	0.0	0.0	25.0	0.0	462.0
Whole cherries	1	17.0	325.5	0.0	6.0	0.0	0.5	0.0	0.0	29.0	0.0	378.0
	2	33.5	434.5	0.5	8.5	0.0	0.5	0.0	0.0	39.5	0.0	517.0
	3	20.5	343.5	0.0	5.5	0.0	0.5	0.0	0.0	27.0	0.0	397.0

**Table 3 antioxidants-14-00995-t003:** Concentration of anthocyanins (μg/g) in different powders.

Sample Name	Biological Replicate Number	Cyanidin-3-glucoside	Cyanidin-3-O-rutinoside	Pelargonidin-3-glucoside	Pelargonidin-3-O-rutinoside	Delphinidinin-3-glucoside	Delphinidinin-3-rutinoside	Petunidin-3-glucoside	Petunidin-3-rutinoside	Peonidin-3-rutinoside	Peonidin-3-glucoside	Total Anthocyanins
Cherry pulp powder (freeze dried)	1	28.0	399.5	0.5	7.0	0.0	0.0	0.0	0.0	25.5	0.0	460.5
	2	28.0	401.5	0.5	7.0	0.0	0.0	0.0	0.0	26.0	0.0	463.0
	3	28.5	401.0	0.5	7.0	0.0	0.0	0.0	0.0	25.0	0.0	462.0
Cherry pulp powder (air dried)	1	2.0	15.5	0.0	0.5	0.0	0.0	0.0	0.0	1.5	0.0	19.5
	2	0.0	1.0	0.0	0.0	0.0	0.0	0.0	0.0	0.0	0.0	1.0
	3	0.5	1.5	0.0	0.0	0.0	0.0	0.0	0.0	0.0	0.0	2.0
Blackberry powder	1	1518.5	307.5	50.5	1.5	82.5	38.5	5.5	1.0	39.5	0.0	2045.0
	2	1414	236.5	33.0	1.0	59.5	29.5	4.0	1.0	26.5	0.0	1807.0
	3	1388	214.5	29.0	1.0	54.0	27.0	3.5	0.5	23.5	0.0	1741.0

## Data Availability

All raw data are available upon request.

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
