# Peer review of "Upcycling of Waste Cherries Produces an Anthocyanin-Rich Powder That Protects Against Amyloid-β Toxicity in C. elegans"

_antioxidants, 2025, doi:10.3390/antiox14080995_

Round 1
Reviewer 1 Report
The study is innovative, the experimental design is solid, and the results are presented with appropriate detail and statistical support.
-
Some figure labels (especially in Figures 2 and 3) are broken across lines and hard to read (e.g., "Ju ice", "Po ma ce"). Please ensure these are clean and consistent for clarity;
-
The discussion appropriately contextualizes the neuroprotective effects of cherry anthocyanins. However, further emphasis on food matrix interactions (e.g., stability, bioaccessibility, gut microbiota interplay) could strengthen the discussion;
-
Consider justifying choosing C. elegans over alternative models, even briefly, in the Introduction or Discussion;
-
Consider also noting the limitations of the study, such as the extrapolation from C. elegans to humans, or lack of bioavailability analysis;
-
Line 302: “...produced from of freeze-dried pulp.” Remove of
-
Line 322: The authors mean prebiotic?
-
Reference #5 and #6 are duplicates. Please revise.
- Reference 12: incomplete
Author Response
We thank the reviewer for their time to review our work and their helpful comments. We have carefully studied the comments and edited the manuscript to address the issues raised. All changes in the manuscript are in blue for the reviewer to easily identify them.
Major comments
Comment 1: The study is innovative, the experimental design is solid, and the results are presented with appropriate detail and statistical support.
Response 1: We thank the reviewer for their constructive feedback, which has helped improve the overall quality of the manuscript.
Detailed comments
Comment 2: Some figure labels (especially in Figures 2 and 3) are broken across lines and hard to read (e.g., "Ju ice", "Po ma ce"). Please ensure these are clean and consistent for clarity;
Response 2: We think this happened during the conversion of the manuscript file from Word to PDF format by the journal. We have now submitted the manuscript as a PDF file to avoid this problem.
Comment 3: The discussion appropriately contextualizes the neuroprotective effects of cherry anthocyanins. However, further emphasis on food matrix interactions (e.g., stability, bioaccessibility, gut microbiota interplay) could strengthen the discussion;
Response 3: We agree with the reviewer that this is important and can be discussed further. In the revised Discussion we have now added:
“Anthocyanins are chemically unstable under neutral or alkaline conditions, and are prone to degradation during gastrointestinal transit. The presence of other components in the cherry pulp matrix, such as fibre pectin and proteins, may act as physical or chemical stabilisers, protecting anthocyanins from early breakdown and enhancing their persistence through the digestive tract. This could help explain the greater biological activity observed with the pulp powder compared to anthocyanin standards or juice alone.”
Comment 4: Consider justifying choosing C. elegans over alternative models, even briefly, in the Introduction or Discussion;
Response 4: We have now included in the Introduction:
“C. elegans offers unique advantages as a whole-organism model for evaluating dietary bioactives, enabling assessment of organism-level responses including immunity, metabolism, and ageing. C. elegans is amenable to rapid, high-throughput screening without carrying the ethical or cost limitations associated with vertebrate models making it ideal for bioactivity screening”. Reference 13 = Corsi et al, 2015.
Comment 5: Consider also noting the limitations of the study, such as the extrapolation from C. elegans to humans, or lack of bioavailability analysis;
Response 5: We agree with the reviewer that the limitations of the study could be made clearer. In the revised Discussion we have added a section on the limitations:
“It is important to note that these results should be interpreted in light of certain limitations. Although C. elegans models conserved molecular pathways, the findings might not directly translate to human biology. Additionally, we did not assess anthocyanin bioavailability, an important factor for determining efficacy.”
Comment 5: Line 302: “...produced from of freeze-dried pulp.” Remove of
Response 5: This has been removed.
Comment 6: Line 322: The authors mean prebiotic?
Response 6: Yes the reviewer is correct of course. This has been changed to “prebiotic”.
Comment 7: Reference #5 and #6 are duplicates. Please revise.
Response 7: This has now been corrected.
Comment 8: Reference 12: incomplete
Response 8: This has now been corrected.

Reviewer 2 Report
In the study, the author demonstrated that upcycling waste cherries produces an anthocyanin-rich powder that protects against amyloid-β toxicity in C. elegans. However, several concerns need to be addressed:
1. Why do cherry juice, pulp, and pomace retain high levels of anthocyanins, particularly Cyanidin-3-O-rutinoside, and also contain quercetin?
2. In Figure 4E, did DMSO have an effect on C. elegans development? If so, what was the reason for this?
3. An earlier study indicated that quercetin protects against amyloid beta-induced toxicity in C. elegans (see doi: 10.1038/s41420-023-01592-x). Why did quercetin not provide protection against amyloid beta-related impairment in this study?
4. Why does cherry pulp protect against amyloid beta toxicity, while the other related compounds tested did not show protective effects?
5. The authors should discuss the limitations of the study
Not applicable
Author Response
We thank the reviewer for their time to review our work and their helpful comments. We have carefully studied the comments and edited the manuscript to address the issues raised. All changes in the manuscript are in blue for the reviewer to easily identify them.
Major comments
In the study, the author demonstrated that upcycling waste cherries produces an anthocyanin-rich powder that protects against amyloid-β toxicity in C. elegans. However, several concerns need to be addressed:
Comment 1: Why do cherry juice, pulp, and pomace retain high levels of anthocyanins, particularly Cyanidin-3-O-rutinoside, and also contain quercetin?
Response 1: We believe the reviewer is asking why these specific bioactive compounds persist across different processing stages and agree this is an important point. The high levels of Cyanidin-3-O-rutinoside and quercetin detected across cherry juice, pulp, and pomace fractions likely reflect the natural distribution of these compounds in cherry fruit and their relative chemical stability. These factors likely contribute to their persistence across fractions. We have added a section in the revised Discussion:
“Our data also show that cherry powder can retain high levels of anthocyanins, in particular Cyanidin-3-O-rutinoside. Cyanidin-3-O-rutinoside is concentrated in the skin and outer mesocarp, tissues that remain in all fractions during processing and is glycosylated, improving its ability to withstand processing steps like juicing or freeze-drying. These findings support the potential of cherry waste-derived powders as a viable approach for retaining stable, bioactive compounds, and their value in developing functional food ingredients.” References: Castañeda-Ovando et al, 2009 and Vidana Gamage et al, 2022.
Comment2: In Figure 4E, did DMSO have an effect on C. elegans development? If so, what was the reason for this?
Response 2: As shown in Fig. 4E and F, C. elegans exposed to DMSO alone exhibited a modest developmental delay. This effect is consistent with previous studies reporting that DMSO at concentrations ≥0.5–1% can induce mild toxicity, including growth delay and reduced locomotion. The cherry products were compared with DMSO control. In the revised Results section of the manuscript we clarify the effect of DMSO:
“Consistent with other studies [21], the solvent control delayed development. After 72 hours ~60% of C. elegans treated with the solvent reached adulthood, compared to ~80% of animals on OP50 only (Fig 4E, F).”
Comment 3: An earlier study indicated that quercetin protects against amyloid beta-induced toxicity in C. elegans (see doi: 10.1038/s41420-023-01592-x). Why did quercetin not provide protection against amyloid beta-related impairment in this study?
Response 3: We agree with the reviewer that the Schiavi et al study is interesting and relevant. We can’t say for sure why different results were obtained, but we note that in Schiavi et al quercetin was mixed with the bacteria before seeding plates, while we added the quercetin stock solution on top of plates that were already seeded with bacteria (and that had formed a lawn for 48 h). As chemical compounds at high concentrations can alter bacterial metabolism, we cannot exclude that the two different protocols lead to different effects on the bacteria, which in turn result in different C. elegans phenotypes. As methods in both manuscripts are/will be clear and transparent, readers can assess this on a case by case basis.
Comment 4: Why does cherry pulp protect against amyloid beta toxicity, while the other related compounds tested did not show protective effects?
Response 4: We agree with the reviewer that this is an important point that can be better explored in the manuscript. Based on our findings, we speculate the matrix of the cherry pulp provides protective effects by e.g. increasing stability or bioaccessibility of anthocyanins. We have now added a section to the revised manuscript further discussing this point [refer also to Reviewer 1 comment 3]:
“Anthocyanins are chemically unstable under neutral or alkaline conditions, and are prone to degradation during gastrointestinal transit. The presence of other components in the cherry pulp matrix, such as fibre pectin and proteins, may act as physical or chemical stabilisers, protecting anthocyanins from early breakdown and enhancing their persistence through the digestive tract. This could help explain the greater biological activity observed with the pulp powder compared to anthocyanin standards or juice alone.”
Comment 5: The authors should discuss the limitations of the study
Response 5: We agree with the reviewer that the limitations of the study could be made clearer. A similar point was also raised by Reviewer 1 [comment 5]. In the revised Discussion we have now added a section on the limitations:
“It is important to note that these results should be interpreted in light of certain limitations. Although C. elegans models conserved molecular pathways, the findings might not directly translate to human biology. Additionally, we did not assess anthocyanin bioavailability, an important factor for determining efficacy.”
Reviewer 3 Report
This manuscript presents a comprehensive analysis of antioxidant compounds, particularly anthocyanins and quercetin, in various forms of waste products from cherry processing. Notably, significant amounts of these compounds were detected in cherry pulp powder. To evaluate the potential application of cherry wastes as a health-promoting food additive, the authors examined their effects on bacteria and C elegans. Interestingly, cherry pulp powder significantly rescued the Aβ-induced paralysis phenotype in C. elegans, a model for Alzheimer’s disease. This finding suggests a potential application of agricultural wastes in the development of foods with protective roles. The experiments were well-designed, the results were carefully analyzed, and the manuscript is well-written. The research aligns well with the scope of this journal.
Major Concern:
Figure 4: In contrast to the protective effect of cherry pulp powder on Aβ-C. elegans, neither cherry juice nor blackberry extract demonstrated similar benefits. Given that all three samples contain considerable levels of anthocyanins and quercetin, this result cannot be simply explained by the antioxidant properties of these compounds. It raises the possibility that other constituents in the pulp powder, or interactions that enhance the bioactivity of anthocyanins and quercetin, may contribute the observed protective effects. Therefore, additional mechanistic studies are expected to provide deeper insight. For example, to determine whether the protective effects are mediated via an anthocyanin-induced oxidative stress response, the authors could assess ROS levels in C. elegans using immunostaining and check the expression of the key oxidative stress-responsive transcription factor SKN-1 (the C. elegans ortholog of Nrf2) by RT-qPCR.
Major Concern:
Figure 4. In contrast to the protective effect of cherry pulp powder on Aβ-C. elegans, neither cherry juice nor blackberry extract demonstrated similar benefits. Given that all three samples contain considerable levels of anthocyanins and quercetin, this result cannot be simply explained by the antioxidant properties of these compounds. It raises the possibility that other constituents in the pulp powder, or interactions that enhance the bioactivity of anthocyanins and quercetin, may contribute the observed protective effects. Therefore, additional mechanistic studies are expected to provide deeper insight. For example, to determine whether the protective effects are mediated via an anthocyanin-induced oxidative stress response, the authors could assess ROS levels in C. elegans using immunostaining and check the expression of the key oxidative stress-responsive transcription factor SKN-1 (the C. elegans ortholog of Nrf2) by RT-qPCR.
Minor Concerns:
Figure 2D: Why do quercetin levels show such high variation in cherry juice samples from different years? A discussion of possible causes would be helpful.
Figure 2B: Why does the pulp powder contain anthocyanin levels approximately 100-fold higher than fresh pulp? Is this due to concentration effects during processing? Please clarify how many grams of fresh pulp are required to produce one gram of pulp powder.
Author Response
We thank the reviewer for their time to review our work and their helpful comments. We have carefully studied the comments and edited the manuscript to address the issues raised. All changes in the manuscript are in blue for the reviewer to easily identify them.
Major comments
This manuscript presents a comprehensive analysis of antioxidant compounds, particularly anthocyanins and quercetin, in various forms of waste products from cherry processing. Notably, significant amounts of these compounds were detected in cherry pulp powder. To evaluate the potential application of cherry wastes as a health-promoting food additive, the authors examined their effects on bacteria and C elegans. Interestingly, cherry pulp powder significantly rescued the Aβ-induced paralysis phenotype in C. elegans, a model for Alzheimer’s disease. This finding suggests a potential application of agricultural wastes in the development of foods with protective roles. The experiments were well-designed, the results were carefully analyzed, and the manuscript is well-written. The research aligns well with the scope of this journal.
Major Concern:
Comment 1: Figure 4: In contrast to the protective effect of cherry pulp powder on Aβ-C. elegans, neither cherry juice nor blackberry extract demonstrated similar benefits. Given that all three samples contain considerable levels of anthocyanins and quercetin, this result cannot be simply explained by the antioxidant properties of these compounds. It raises the possibility that other constituents in the pulp powder, or interactions that enhance the bioactivity of anthocyanins and quercetin, may contribute the observed protective effects. Therefore, additional mechanistic studies are expected to provide deeper insight. For example, to determine whether the protective effects are mediated via an anthocyanin-induced oxidative stress response, the authors could assess ROS levels in C. elegans using immunostaining and check the expression of the key oxidative stress-responsive transcription factor SKN-1 (the C. elegans ortholog of Nrf2) by RT-qPCR.
Response 1: We agree that evaluating the role of oxidative stress pathways would be important for understanding the mechanism behind the protective effects of cherry pulp powder. Rather than immunostaining and RT-qPCR, which are lower-throughput and more variable in C. elegans, utilised a well-established SKN-1-responsive reporter strain: gst-4p::GFP. This strain enables sensitive, in vivo monitoring of SKN-1 activation in response to oxidative stress and dietary interventions, and has been widely used to probe oxidative stress pathway activation. We found that neither cherry juice, blackberry powder, nor cherry pulp powder alters gst-4::GFP levels relative to vehicle control. In contrast, 1 mM paraquat, a known pro-oxidant, robustly increased gst-4::GFP levels, confirming the assay's responsiveness. These results suggest that the protective effects of cherry pulp powder in the Aβ model are unlikely to be mediated through SKN-1-driven oxidative stress response pathways. We have now included these data and the interpretation in the Results section of the revised manuscript (see below). The appropriate Methods are also added.
“Our data showing that the cherry pulp powder, but not the cherry juice, blackberry powder, individual anthocyanins nor quercetin, exhibited protective effects were unexpected since all products contain high levels of anthocyanins. These findings raise the possibility that the observed protection against proteotoxicity may not be mediated via an anthocyanin-induced oxidative stress response. To examine the effect of the products on oxidative stress pathway activation, we utilised a well-established Nrf-2/SKN-1-responsive reporter strain, gst-4p::GFP which enables in vivo monitoring of Nrf-2/SKN-1 activation in response to oxidative stress and dietary interventions using fluorescence microscopy. We found that neither cherry juice, blackberry powder, nor cherry pulp powder alters gst-4::GFP levels relative to vehicle control. In contrast, 1 mM paraquat, a known pro-oxidant, robustly increased gst-4::GFP levels, confirming the assay's responsiveness (Fig S3A, B). These results suggest that the protective effects of cherry pulp powder in the Aβ model are unlikely to be mediated through Nrf2/SKN-1-driven oxidative stress response pathways. A possibility is that other constituents in the pulp powder, or interactions that enhance the bioactivity of anthocyanins and quercetin, may contribute the observed protective effects, and that more complex interactions between bioactive compounds in a food matrix is required to improve health.”
Minor Concerns:
Comment 2: Figure 2D: Why do quercetin levels show such high variation in cherry juice samples from different years? A discussion of possible causes would be helpful.
Response 2: We agree with the reviewer that this is a point worth of discussion. We have added the following to the revised Results:
“Our analyses showed a high degree of variation in quercetin across cherry juice batches. This likely reflects year-to-year differences in growing conditions such as sunlight, temperature, and water availability, all of which are known to influence flavonoid biosynthesis” references 15 and 16 (Enaru et al, 2021 and UK Met climate summary).”
Comment 3: Figure 2B: Why does the pulp powder contain anthocyanin levels approximately 100-fold higher than fresh pulp? Is this due to concentration effects during processing? Please clarify how many grams of fresh pulp are required to produce one gram of pulp powder.
Response 3: There are indeed concentrations effects due to the processing. These numbers are shown in Fig 1B. 3 kg of fresh cherries are used to generate 1 L of juice while 250 grams pulp (frozen) generates 50-100 grams of powder. Thus the powder is highly concentrated in terms of fruit content.
Round 2
Reviewer 2 Report
N/A
N/A
Reviewer 3 Report
The revised manuscript has addressed all the reviewer comments. In particular, in response to the major concern, the authors have added a mechanistic study using a reporter system to demonstrate that the protective effects observed are unlikely to be mediated by the classic oxidative stress response pathways. The current version of the manuscript is well-suited for publication.
The revised manuscript has addressed all the reviewer comments. In particular, in response to the major concern, the authors have added a mechanistic study using a reporter system to demonstrate that the protective effects observed are unlikely to be mediated by the classic oxidative stress response pathways. The current version of the manuscript is well-suited for publication.